# Comparative Analyses of Production Performance, Meat Quality, and Gut Microbial Composition between Two Chinese Goose Breeds

**DOI:** 10.3390/ani12141815

**Published:** 2022-07-15

**Authors:** Hongyu Ni, Yonghong Zhang, Yuwei Yang, Yumei Li, Yijing Yin, Xueqi Sun, Hengli Xie, Jinlei Zheng, Liping Dong, Jizhe Diao, Hao Sun, Yan Zhang, Shuang Liang

**Affiliations:** 1Department of Animals Sciences, College of Animal Science, Jilin University, Changchun 130062, China; nihy20@mails.jlu.edu.cn (H.N.); yonghong@jlu.edu.cn (Y.Z.); yangyuwei@jlu.edu.cn (Y.Y.); li_ym@jlu.edu.cn (Y.L.); yinyj21@mails.jlu.edu.cn (Y.Y.); xiehl19@mails.jlu.edu.cn (H.X.); zhengjl20@mails.jlu.edu.cn (J.Z.); donglp20@mails.jlu.edu.cn (L.D.); diaojz20@mails.jlu.edu.cn (J.D.); sunhao92@jlu.edu.cn (H.S.); 2Jilin Academy of Agricultural Sciences, Changchun 130033, China; bc_lky@163.com; 3College of Animal Science and Technology, Jilin Agriculture Science and Technology University, Jilin 132109, China

**Keywords:** abdominal fat, muscle yield, fatty acids, cecal microbiota

## Abstract

**Simple Summary:**

Poultry is one of the most frequently consumed meats in the world and plays an important role in the daily life of people. Goose meat is consumed by consumers because it contains a relatively high proportion of polyunsaturated fatty acids. Meat quality traits, production performance, and cecal microbiota diversity in two goose breeds (Zi goose and Xianghai flying goose) were evaluated in this study. Understanding these aspects not only provides a reference for the exploration of the relationship between the cecal microbiota and production performance but also guidelines for the human consumption of healthy poultry meat.

**Abstract:**

Goose meat is consumed by consumers because it contains a relatively high proportion of polyunsaturated fatty acids (PUFAs). This study was conducted to explore the main differences in production performance, breast meat quality traits, and cecal microbiota compositions between the Zi goose (ZG) and Xianghai flying goose (FG). The production performance and breast meat quality trait analyses showed that compared with the ZG, the FG had a higher right breast muscle index, ileum villi height/crypt depth ratio (VH/CD), and cecum fermentation rate (higher short-chain fatty acid (SFCA) concentration); a lower abdominal fat index; a higher proportion of PUFAs; and a lower shear force. Spearman’s correlation coefficients between the cecal microbiota composition and production performance indexes suggested that the genus *Faecalibacterium* was positively associated with production performance; in contrast, the genus *Candidatus Saccharimonas* was negatively correlated with production performance; moreover, the *Ruminococcus torques group*, *Parasutterella*, and *Methanobrevibacter* were negatively related to the VH/CD. Taken together, in this particular trial, FG had better production performance, healthier meat quality traits, and better intestinal digestion and absorption capacities than ZG. These results not only provide a useful data reference for the production of healthy geese for human consumption but can also help guide the utilization of goose breed resources.

## 1. Introduction

Poultry is one of the most frequently consumed meats in the world and plays an important role in the daily life of people. Therefore, the main goal of commercial poultry breeding farms is to achieve faster growth and increased breast muscle yield in birds. However, a rapid increase in the growth rate of poultry is usually accompanied by an increase in abdominal fat accumulation during poultry farming [1]. Abdominal fat is considered a “waste product” with little commercial value [2]. In addition, excessive abdominal fat deposition can reduce feed conversion efficiency and thus increase feed costs [1]. Intramuscular fat (IMF) content affects the accumulation of fatty acids and the development of flavor, which is highly correlated with consumer preferences [3]. However, an increasing number of consumers are concerned that consuming meat with too much fat may lead to increased risks of cardiovascular disease (CVD), atherosclerosis, and obesity. These concerns are based on a previous study reporting that saturated fat in meat has negative health effects [4], as saturated fatty acids (SFAs) can increase plasma cholesterol levels, which is a potential risk factor for atherosclerosis [5]. In contrast, polyunsaturated fatty acids (PUFAs) are beneficial to human health because of their ability to reduce the risk of atherosclerosis and thrombosis [6]. Therefore, with the improvement of living standards, consumers are increasingly interested in organic poultry meat with high levels of PUFAs.

The goose (*Anas cygnoides*) is a traditional herbivorous poultry animal with meat rich in PUFAs that is favored by consumers. In recent years, the intestinal tract microbial community has attracted increasing attention for its important role in improving growth performance, aiding nutrient absorption, regulating host physiology, maintaining host health, and ensuring the safety of animal products [7,8]. In poultry, the intestine is the main site of microbial colonization. Studies have shown that the cecal microbiota of poultry is closely associated with the host and ingested feed [9]. Hence, cecal microbiota analysis is a key field of poultry nutrition research. Although many studies have examined the effects of breed on cecal microbial community structure, there is limited data from goose models. The Zi goose (ZG) and Xianghai flying goose (FG) are two native goose breeds in China that are mainly distributed in northeast China (Figure 1). ZG is famous for its high egg production, and FG is known for its characteristics of strong flying ability, rough feed tolerance, and strong disease resistance. However, little is known about the cecal microbial communities in these two species.

Therefore, the objectives of this study were as follows: (1) to identify the major differences in production performance and meat quality traits between the ZG and FG; (2) to identify and compare the microbiota composition between the ZG and FG; and (3) to elucidate how the cecal microbial community structure affects production performance in the ZG and FG. Understanding these aspects not only provides a reference for exploration of the relationship between the cecal microbiota and production performance but also provides theoretical support for the utilization of goose breed resources and guidelines for the human consumption of healthy poultry meat.

## 2. Materials and Methods

### 2.1. Animal Experiments

A total of sixty 1-day-old healthy goslings were sampled. ZG (82.45 ± 1.39 g; *n* = 30) and FG (82.97 ± 7.20 g; *n* = 30) individuals were provided by Jiuzhou Flying Goose Husbandry & Technology Co., Ltd., Baicheng, China, and raised on the Laboratory Animal Farm of Jilin University, China, from 7 June 2021 to 25 October 2021. Sixty goslings were kept in a shed to ensure stable rearing conditions and then gradually transferred to the outside. The first day of breeding was under 24 h of light, which gradually decreased to natural light conditions in week 8. The indoor temperature ranged within 25–26 °C in the first week of breeding, then decreased to 20 °C in week 4 (approximately 2 °C a week), and finally was the same as the ambient temperature in week 8. The two breeds of geese were raised in two separate pens (20 m × 30 m), with 30 birds per pen, for a total of 2 pens, and each pen was divided into 5 small pens with 6 geese in each small pen. A commercial gosling diet and normal growth goose commercial diet (Table 1) were offered ad libitum, and water was available throughout the whole trial. All the birds were routinely immunized using the neck subcutaneous injection method according to the vaccine immunization schedule but did not receive antibiotic or probiotic treatment.

### 2.2. Sample Collection

Bird body weight (BW) was measured after fasting for 12 h on day 140 of the experiment, and then the birds were sacrificed by cervical dislocation. Abdominal fat mass and the right breast muscle were weighed immediately after scalding and plucking to calculate abdominal fat and right breast muscle indexes. Cecum contents were collected in sterile 5 mL polypropylene tubes and frozen at −80 °C for further DNA extraction and short-chain fatty acid (SFCA) analysis. Thereafter, a portion of the middle ileum (1 cm piece; washed with PBS) and the left breast muscle (0.5 cm × 0.5 cm × 1.0 cm pieces) were collected and stored in 4% formaldehyde-phosphate buffer for morphological analysis. The remaining left breast muscle samples were divided into two parts: one part was removed from the refrigerator and preserved at −20 °C until fatty acid analysis, while the other part was placed on ice, transferred to the laboratory, and kept at 4 °C. After a 24 h aging period, the meat quality traits were analyzed immediately.

### 2.3. Morphologic Examination of Ileum and Muscle Fiber Tissue

To evaluate the ileum and muscle fiber morphological characteristics, the ileum and muscle sections were stained with hematoxylin and eosin (H&E) according to a previous study [10], with a slight modification. In brief, each ileum and muscle sample was fixed in 4% formaldehyde-phosphate buffer (Solarbio, Beijing, China) overnight, embedded in paraffin, cut into 5-μm-thick sections, and stained with H&E (Solarbio, Beijing, China) for morphological analysis. The villus height and crypt depth of ileum samples at a magnification of 40× and the fiber diameter, cross-sectional area (CSA), and density of muscle samples at a magnification of 400× (a total of approximately 300 muscle fibers for each muscle sample) were calculated using Image-Pro Plus 6.0 software (Nikon, Tokyo, Japan).

### 2.4. Meat Quality Trait Analysis

#### 2.4.1. Physical Properties

The meat color and pH values of the breast muscle samples were measured at three different locations by a carcass color tester (OPTO-STAR, Beijing Bulader Technology Development Co., Ltd., Denmark, Germany) and portable pH meter (pH-STAR, Beijing Bulader Technology Development Co., Ltd., Denmark, Germany) according to the instructions. The pH meter was calibrated with standard buffers with pH values of 4.0 and 7.0. Cooking loss and shear force [11,12] were determined as described in previous studies, with slight modifications. In brief, breast muscle samples were packaged in cooking bags and placed in a water bath at 80 °C until the central temperature reached 70 °C. After cooling, the samples were weighed again to calculate the cooking loss. Subsequently, these muscle samples were used for the shear force analysis. In brief, cores with a diameter of approximately 1 cm were cut parallel to the muscle fiber orientation at different positions of breast muscle, and shear force was measured using a digital meat tenderness instrument (C-LM3B, Tenovo, Beijing, China). Water loss was analyzed with a digital dilatometer (C-LM3B, Tenovo, Beijing, China) as described by Huo et al. [13]. Approximately 1 g (W_1_) of muscle was weighed, and 10 layers of filter paper were placed on the top and bottom of the sample. Then, the covered sample was placed on the dilatometer platform for 5 min at a pressure of 68.66 kPa, and the weight of the muscle sample was measured again (W_2_) to calculate the amount of released water as follows: Water loss (%) = (W_1_ − W_2_)/W_1_ × 100%.

#### 2.4.2. Proximate Composition

The IMF content of breast muscle was determined by the Soxhlet extraction method with anhydrous ether as the extraction solvent [14] and expressed as the weight percentage of dry matter muscle tissue. Breast muscle fatty acid amounts were analyzed through transmethylation of the fatty acids and quantification as described in previous research [15] with minor modifications. In brief, 2 g of ground breast meat sample and 200 µL of internal standard (C13 in n-hexane) in 15 mL of Folch solution (chloroform: methanol = 2:1) were mixed thoroughly by vortexing. After overnight equilibration at 4 °C, 5 mL of 0.74% NaCl was added, mixed, and then centrifuged at 3000 rpm for 15 min to separate the solvent layers. The bottom layer containing the fatty acid methyl esters (FAMEs) was analyzed on a gas chromatograph (GC) equipped with a Restek RTX 2330 column (105 m × 0.21 mm × 0.20 µm film thickness, Restek Corporation, Bellefonte, PA, USA) and a flame ionization detector. The temperature of the oven was set at 175 °C for 17 min, programmed to 220 °C at 6 °C per min, and maintained at 220 °C for 10 min. The carrier gas was high-purity hydrogen with a flow rate of 50 cm/s and a split ratio of 80:1. The injection and detector temperatures were set to 260 °C and 300 °C, respectively. Equations were generated for response and conversion factors to quantify individual fatty acids from FAMEs and quantified using the internal standard calibration method.

### 2.5. DNA Extraction, Microbiota Analysis, and Functional Prediction

The total microbial genomic DNA of the cecal content samples was extracted using a Magnetic Soil and Stool DNA Kit (Tiangen Biotech Co., Ltd., Beijing, China) according to the manufacturer’s protocol. DNA quantity and concentration were determined by a Quant-iT PicoGreen dsDNA Assay Kit (Invitrogen, Carlsbad, CA, USA) and NanoDrop 1000 spectrophotometer (Nanodrop Technologies, Wilmington, DE, USA). The bacterial 16S rRNA hypervariable V3-V4 region was amplified using 341F CCTAYGGGRBGCASCAG and 806R GGACTACNNGGGTATCTAAT primers [16]. The 250-bp paired-end amplicon libraries were sequenced using the Illumina NovaSeq 6000 platform (Illumina, San Diego, CA, USA). The amplicon sequence data were processed with QIIME2 software package (https://qiime2.org, accessed on 9 May 2022) [17]. Briefly, the amplicon paired sequences were demultiplexed with the demux plugin followed by primer trimming off using the cutadapt plugin. Then, the DADA2 plugin was used to denoise by filtering out low-quality sequences with a Q < 20 and merge high-quality paired-end clean sequences into tags followed by removing chimera and singletons [18]. Nonsingleton amplicon sequencing variants (ASVs) were classified into taxa according to the Silva database (http://www.arb-silva.de, accessed on 9 May 2022). The alpha diversity (Chao1, Shannon, and Simpson indexes) of each sample was analyzed based on the rarefied ASVs. Beta diversity was examined based on the Bray–Curtis distance and displayed by principal coordinate analysis (PCoA) in the R language (http://www.r-project.org/, accessed on 9 May 2022). A heatmap was generated based on phylum and genus information using the R heatmap plugin (http://www.r-project.org/, accessed on 9 May 2022). The differentially abundant taxa between ZG and FG were determined based on the Benjamini–Hochberg corrected *p* value (false discovery rate < 0.05) using Metastats software (http://metastats.cbcb.umd.edu/, accessed on 20 May 2022).

### 2.6. Short-Chain Fatty Acid Analysis

The SCFA concentrations of the cecal digesta were determined as described previously, with slight modification [19], using a GC equipped with an Agilent DB-5 column (30 m × 0.25 mm × 0.25 μm film thickness, Santa Clara, CA, USA). The initial oven temperature was set at 10 °C for 2 min, programmed to 200 °C at 15 °C per min, and held for 5 min. The injection temperature was 260 °C, the carrier gas was high-purity nitrogen with a flow rate of 25 mL per min and a split ratio of 25:1. The runtime for each analysis was 12.95 min.

### 2.7. Statistical Analysis

The relationship between the cecal microbiota composition and the production performance indexes, including abdominal fat index, right breast muscle index, villus height/crypt depth ratio (VH/CD), and total SCFAs, were investigated by using the R package (Version 2.15.3) to evaluate Spearman’s correlation coefficients. Statistical significance was assessed with an unpaired, two-tailed Student’s t test or Mann–Whitney test in GraphPad Prism software (version 8, GraphPad Software Inc., San Diego, CA, United States). Data are presented as the mean ± SEM, with * *p* < 0.05, ** *p* < 0.01, *** *p* < 0.001, and **** *p* < 0.0001.

## 3. Results

### 3.1. Comparisons of Production Performance and Ileum Epithelial Histological Characteristics between the Two Goose Breeds

At the end of the trial, the FG had a significantly lower BW (3.22 ± 0.17 kg; *p* = 0.008) and abdominal fat index (22.29 ± 1.29 g/kg; *p* = 0.011) but a higher right breast muscle index (65.76 ± 1.76 g/kg; *p* < 0.001) than the ZG (BW, 3.76 ± 0.09 kg; abdominal fat index, 26.92 ± 0.92 g/kg; right breast muscle index, 56.76 ± 1.49 g; Figure 2A,C,D). There was no difference (*p* > 0.137) observed in the right breast muscle weight between the ZG (205.49 ± 4.32 g) and FG (219.85 ± 7.65 g; Figure 2B). The histological analyses of the ileum showed that although both the ZG and FG had healthy intestinal mucosal barrier function, FG had a higher villus height (705.14 ± 11.19 μm; *p* = 0.001), lower crypt depth (164.52 ± 9.41 μm; *p* = 0.003), and higher VH/CD (4.42 ± 0.18; *p* < 0.0001) than ZG (villus height, 634.99 ± 12.17 μm; crypt depth, 232.02 ± 15.05 μm; VH/CD, 2.91 ± 0.15; Figure 2E,F).

### 3.2. Comparison of Breast Muscle Quality Traits between the Two Goose Breeds

#### 3.2.1. Physical Properties and Muscle Fiber Characteristics

The physical properties and muscle fiber characteristics of breast muscle samples are listed in Table 2 and Figure 2G,H. The meat color (*p* = 0.010), water loss (*p* = 0.002), and muscle fiber density (*p* = 0.040) in FG were greater than those in ZG; the pH (*p* = 0.003), shear force (*p* = 0.047), CSA (*p* = 0.043), and muscle fiber diameter (*p* = 0.034) were significantly lower in FG (muscle fiber density, 1995.50 ± 192.72 number/mm^2^; CSA, 517.50 ± 46.69 μm^2^; muscle fiber diameter, 12.78 ± 0.59 μm) than in ZG (muscle fiber density, 1198.39 ± 99.13 number/mm^2^; CSA, 915.05 ± 100.42 μm^2^; muscle fiber diameter, 16.84 ± 0.87 μm). Meanwhile, there was no difference in the cooking loss between ZG and FG (*p* = 0.063).

#### 3.2.2. Proximate Composition

The IMF contents and fatty acid proportions in ZG and FG are presented in Table 3. FG had a significantly lower relative proportion of IMF than ZG (*p* = 0.001), which was consistent with abdominal fat deposition, suggesting that a rapid increase in abdominal fat deposition is accompanied by an increase in IMF deposition. Oleic acid (C18:1n9c) was the most abundant fatty acid in both ZG and FG, followed by palmitic acid (C16:0) and stearic acid (C18:0). The proportions of C18:0, linoleic acid (C18:2n6c), arachidonic acid (C20:4n6), docosahexaenoic acid (C22:6n3), and PUFAs were markedly higher in the FG than in the ZG (*p* < 0.05). The proportions of C16:0, C18:1n9c, and monounsaturated fatty acids (MUFAs) were significantly higher in the ZG than in FG (*p* < 0.05). Moreover, there was no significant difference in the proportions of tetradecanoic acid (C14:0), palmitoleic acid (C16:1), α-linolenic acid (C18:3n3), cis-8,11,14-eicosenotrienoic acid (C20:3n6), and unsaturated fatty acids (SFAs) between ZG and FG (*p* > 0.05).

### 3.3. Comparison of Cecum Microbiota Composition, Functional Prediction, and SCFA Concentrations between the Two Goose Breeds

An average Good’s coverage of 100% was observed for both ZG and FG samples. No differences were observed in the Observed ASVs (*p* > 0.999), Chao1 (*p* > 0.999), Shannon (*p* = 0.310), and Simpson (*p* = 0.087) between the ZG and FG (Table 4). However, principal coordinates analysis (PCoA) and Bray–Curtis dissimilarity distance analysis (Figure 3A,B) showed significant differences in the microbial composition of the cecum in the ZG compared with the FG (*p* < 0.05). Specifically, FG had a significantly higher abundance of *Faecalibacterium* (*p* = 0.026) and *Intestinimonas* (*p* = 0.024) and a lower abundance of *Parabacteroides* (*p* = 0.020), *Ruminococcus torques group* (*p* = 0.010), *Clostridia UCG-014* (*p* = 0.010)*, Methanobrevibacter* (*p* = 0.008)*, Parasutterella* (*p* = 0.037)*, Candidatus Saccharimonas* (*p* = 0.009), and *Streptococcus* (*p* = 0.034) than ZG (Figure 3C,D and Figure 4). SCFA concentration analysis is presented in Figure 5. The most abundant SCFA was acetic acid, followed by propionic, butyric, and isobutyric acids in both bird breeds. In addition, the concentrations of propionic acid (*p* < 0.0001), butyric acid (*p* < 0.0001), isobutyric acid (*p* = 0.042), valeric acid (*p* = 0.012), hexanoic acid (*p* = 0.021), and total SCFAs (*p* = 0.005) in the FG cecum (propionic acid, 104.19 ± 10.06 μg/g; butyric acid, 69.35 ± 6.99 μg/g; isobutyric acid, 9.94 ± 0.99 μg/g; valeric acid, 7.74 ± 1.49 μg/g; hexanoic acid, 5.72 ± 0.90 μg/g; total SCFAs, 1373.76 ± 93.41 μg/g) were significantly higher than those in the ZG cecum (propionic acid, 8.24 ± 0.95 μg/g; butyric acid, 9.60 ± 0.24 μg/g; isobutyric acid, 7.33 ± 0.24 μg/g; valeric acid, 2.65 ± 0.05 μg/g; hexanoic acid, 2.96 ± 0.12 μg/g; total SCFAs, 1072.85 ± 111.29 μg/g). 

### 3.4. Associations of the Cecum Microbiota Composition with Production Performance Indexes

We then investigated the relationships between the cecal microbiota composition and production performance indexes with Spearman’s correlation coefficients (Figure 6). The results showed that *Faecalibacterium* was positively associated with the right breast muscle index (*p* < 0.01), VH/CD (*p* < 0.05), and total SCFAs (*p* < 0.05), while *Candidatus Saccharimonas* was negatively related to the right breast muscle index (*p* < 0.05) and VH/CD (*p* < 0.05). *Candidatus Saccharimonas* had an extremely significant positive correlatio with the abdominal fat index (*p* < 0.01); *Ruminococcus torques group* (*p* < 0.01), *Parasutterella* (*p* < 0.05), and *Methanobrevibacter* (*p* < 0.05) were significantly negatively correlated with VH/CD. 

## 4. Discussion

The demand for geese as a source of meat for human consumption has increased in recent years. The main goal of poultry production is to obtain higher muscle yield. With improvements in living standards, the need for high-quality meat products has greatly increased. Therefore, increasing yield is a key issue for the meat industry. Nevertheless, there are many factors affecting production performance, among which breed is an important factor. Different goose breeds have different gut microbial compositions; conversely, different gut microbial compositions contribute to different production performances. Therefore, meat quality traits, production performance, and cecal microbiota diversity in two goose breeds (ZG and FG) were evaluated in this study.

China is the most productive goose producer in the world, and high meat quality is a major attribute that influences consumer acceptance, which is especially important for the meat industry. Weng et al. [20] indicated that with an increasing CSA, the pH of postmortem breast muscle decreased slowly, and the glycolysis potential of muscle decreased, which eventually led to an increase in the final pH of meat. Additionally, muscle with a larger fiber size exhibited lower drip loss values and a lighter meat color than muscle with a smaller fiber size [21,22]. The results of the current research are consistent with those of previous studies. Generally, muscles with a smaller CSA are considered to be of good meat quality [23], which may be due to the effect of muscle fiber size on fiber bundle size and muscle growth potential, resulting in visible roughness in the cross section of meat. This was confirmed in our findings, given that the goose breast meat with a lower shear force (better tenderness) had a thinner muscle fiber diameter, smaller CSA, and higher muscle fiber density. The amount of IMF is highly correlated with the development of meat flavor. However, some consumers have been concerned that excessive dietary fat intake may increase the risk of CVD or obesity. On the other hand, studies have revealed that the composition of fatty acids has a more profound impact on human health than the amount of fat in the diet [24,25,26]. It was reported that the dietary intake of high SFAs had adverse effects on human health; however, a high intake of PUFAs, especially n-3 fatty acids, was positively associated with reduced risks of some diseases [25]. SFAs from C12:0 to C16:0 are cholesterol-raising fatty acids, while C18:0 is not, mainly because C18:0 is easily desaturated to C18:1n9c [25]. Therefore, it is healthier for consumers to choose meat with a lower proportion of harmful SFAs. In the present research, there was no distinct difference in the SFA proportion between the ZG and FG meats; however, meat from the FG had a significantly lower proportion of C16:0 than meat from the ZG. A lower IMF content but higher amounts of PUFAs and C22:6n3 (n-3 fatty acid) were also found in FG meat. These results indicated that although ZG meat may be more popular with consumers in terms of flavor, FG meat products are of better quality and healthier for consumers to eat.

The cecum comprises a complex ecosystem consisting of a highly diverse microbiome. Alpha diversity generally evaluates the microbial community diversity of a single sample, while beta diversity is used to reflect differences in the species complexity of samples at the group level. In the present study, there was no significant difference in the alpha diversity of the cecal microbiota between the ZG and FG. However, the Bray–Curtis distance, representative of beta diversity, showed that the cecal microbiota composition of the FG was distinctly different from that of the ZG, suggesting that gut development and key phylotypes of the cecal microbiota were significantly different between the ZG and FG. It was reported that the cecal digestion mechanism in geese is similar to that of a rumen [8]. Some cellulose that is difficult to digest by host enzymes, such as plant cellulose, resistant starch, and oligosaccharides, can be fermented by cecal microorganisms to produce SCFAs that are absorbed by the intestinal epithelium as additional energy for the host [27]. *Ruminococcaceae* are mainly cellulolytic microorganism species that can degrade cellulose effectively [28], and *Intestinimonas* are butyrate-producing bacteria [29,30]. Here, we found higher abundances of *Faecalibacterium* (family *Ruminococcaceae*) and *Intestinimonas* in the cecal contents of the FG than in those of the ZG. Furthermore, the ileum VH/CD was also higher in the FG than in the ZG. Intestinal nutrient absorption in poultry is mainly carried out in the small intestine. A higher VH/CD represents a higher intestinal nutrient absorption capacity [31]. These results suggested that FG has excellent digestive and nutrient absorption abilities compared with ZG. Therefore, FG had a higher cecal fermentation ability (higher concentrations of propionic acid, butyric acid, isobutyric acid, valeric acid, hexanoic acid, and total SCFAs) and higher muscle yield (higher right breast muscle index) than ZG. Moreover, the relative abundances of *Parabacteroides*, *Ruminococcus torques group*, *Clostridia UCG-014*, *Methanobrevibacter*, *Parasutterella, Candidatus Saccharimonas*, and *Streptococcus* were significantly higher in the ZG than in FG. Study has reported that the *Ruminococcus torques group* was significantly correlated with fat deposition in ducks [32]. Therefore, the ZG may have a better fat deposition ability than the FG. Consistent with this finding, the ZG showed a higher abdominal fat index than the FG. Excessive abdominal fat deposits have been proven to reduce feed conversion efficiency [1]. *Methanobrevibacter* is the main methanogen. Higher methane formation represents a higher loss of gross energy intake [33]; moreover, *Candidatus Saccharimonas* is an opportunistic pathogen in the intestinal tract [34], which is associated with gastrointestinal disorders, while *Parabacteroides* [35], *Parasutterella* [36], *Clostridia UCG-014*, and *Streptococcus* [37] were related to promoting gut inflammation. These may be the reasons why a lower ileum VH/CD was found in the ZG than in the FG. Thereafter, we investigated the relationship between the cecal microbiota composition and the production performance indexes in geese by Spearman’s correlation analysis and found that *Faecalibacterium* was positively associated with breast muscle yield, VH/CD, and total SCFAs, whereas *Candidatus Saccharimonas* was negatively correlated with breast muscle yield but positively correlated with abdominal fat index. Moreover, *Ruminococcus torques group*, *Parasutterella,* and *Methanobrevibacter* were negatively associated with VH/CD. These findings indicate that the cecal microbiota plays a key role in the production performance of geese.

## 5. Conclusions

Different goose breeds have diverse cecal microbiota compositions, and various cecal microbiota compositions contribute to dissimilar production performance traits. In this particular trial, the results showed that FG had a lower fat deposition capacity in abdominal adipose tissue, healthier meat quality traits, a higher breast meat yield, and better intestinal microbial digestion and absorption abilities than ZG, which may be related to its strong flying ability, rough feed tolerance, and strong disease resistance characteristics. These results provide not only a useful data reference for the human consumption of healthy goose meat but also theoretical support for the development and utilization of goose breed resources.

## Figures and Tables

**Figure 1 animals-12-01815-f001:**
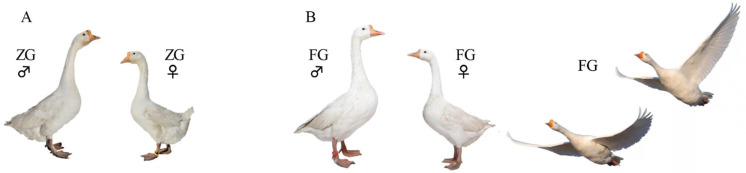
Pictures of (**A**) Zi geese and (**B**) Xianghai flying geese.

**Figure 2 animals-12-01815-f002:**
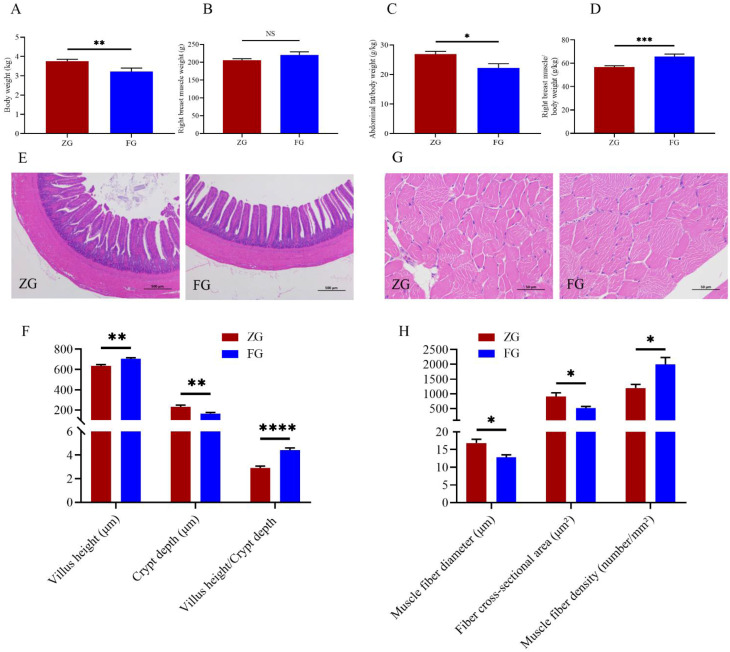
Growth performance and histological characteristics of the two goose breeds. (**A**) Body weight and (**B**) right breast muscle weight were measured on the final day of the experiment; (**C**) abdominal fat index and (**D**) right breast muscle index were defined as the abdominal fat mass and right breast muscle weight divided by the body weight, respectively; *n* = 30. H&E staining of the (**E**) ileum at 40× magnification and (**G**) breast muscle fiber at 400× magnification. Comparison of histological characteristics of the (**F**) ileum (villus height, crypt depth, and VH/CD) and (**H**) breast muscle fiber (muscle fiber diameter, fiber cross-sectional area, and muscle fiber density) of the two goose breeds; *n* = 9. Vertical bars represent the mean ± SEM. NS: No significant; * *p* < 0.05, ** *p* < 0.01, *** *p* < 0.001, and **** *p <* 0.0001 indicate significant differences between the ZG and FG. ZG, Zi goose; FG, Xianghai flying goose.

**Figure 3 animals-12-01815-f003:**
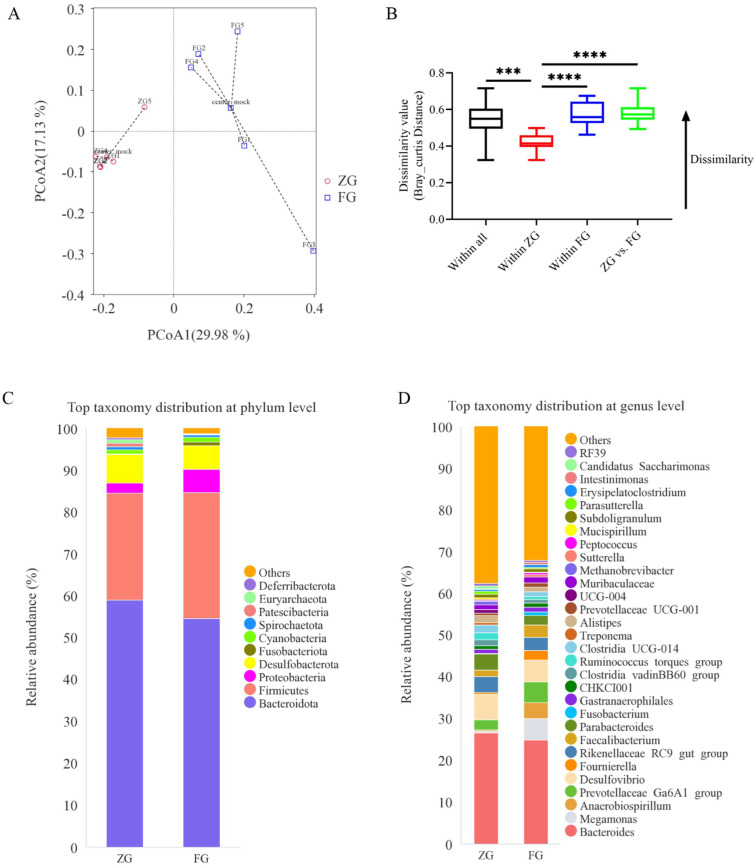
Differences in the cecal microbial composition between the two goose breeds. (**A**) Principal coordinate analysis (PCoA) plot showing cecal microbial compositional differences as quantified by (**B**) Bray–Curtis distance; (**C**,**D**) taxonomic compositions; *n* = 5. *** *p* < 0.001 and **** *p <* 0.0001 indicate significant differences between the ZG and FG. ZG, Zi goose; FG, Xianghai flying goose.

**Figure 4 animals-12-01815-f004:**
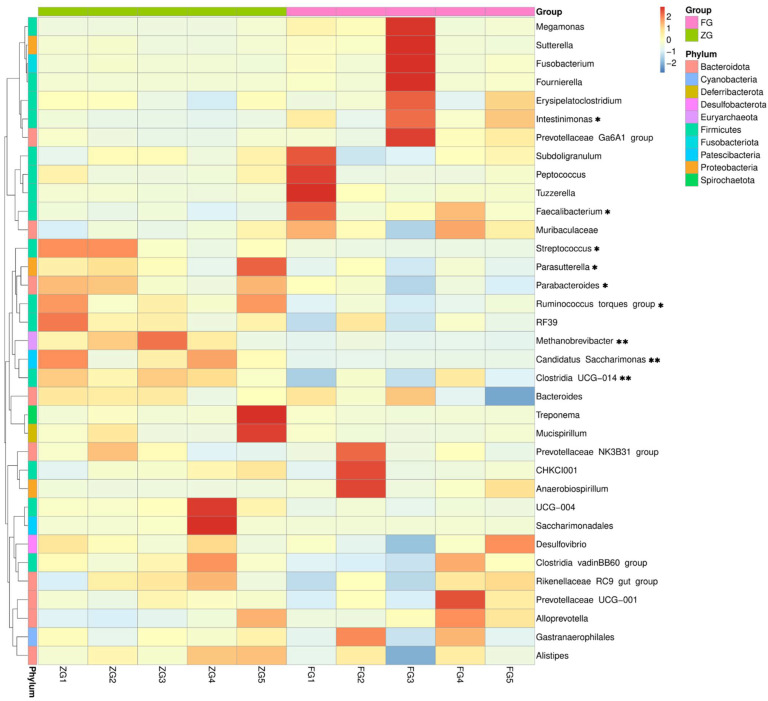
Heatmap of major genera of cecal microbiota. * *p* < 0.05 and ** *p* < 0.01 indicate significant differences between the ZG and FG. ZG, Zi goose; FG, Xianghai flying goose.

**Figure 5 animals-12-01815-f005:**
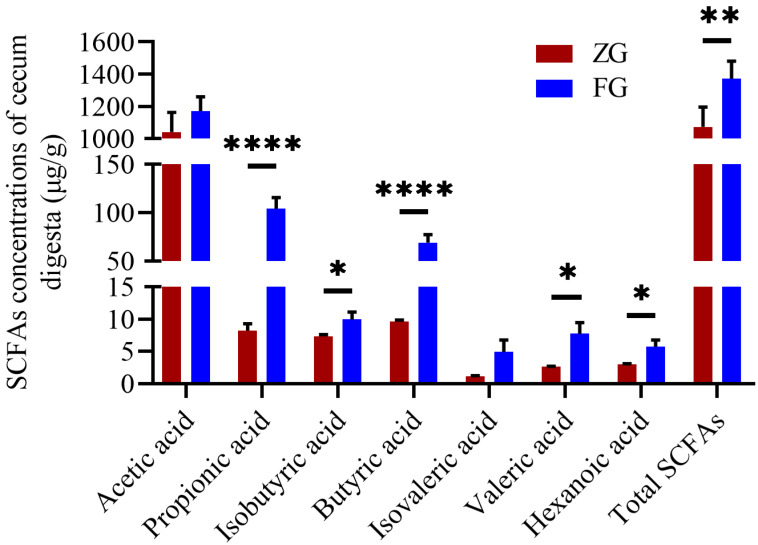
Differences in cecum total SCFA concentrations between the two goose breeds; *n* = 5. Vertical bars represent the mean ± SEM. * *p* < 0.05, ** *p* < 0.01, and **** *p <* 0.0001 indicate significant differences between the ZG and FG. ZG, Zi goose; FG, Xianghai flying goose; Total SCFAs, total short-chain fatty acids.

**Figure 6 animals-12-01815-f006:**
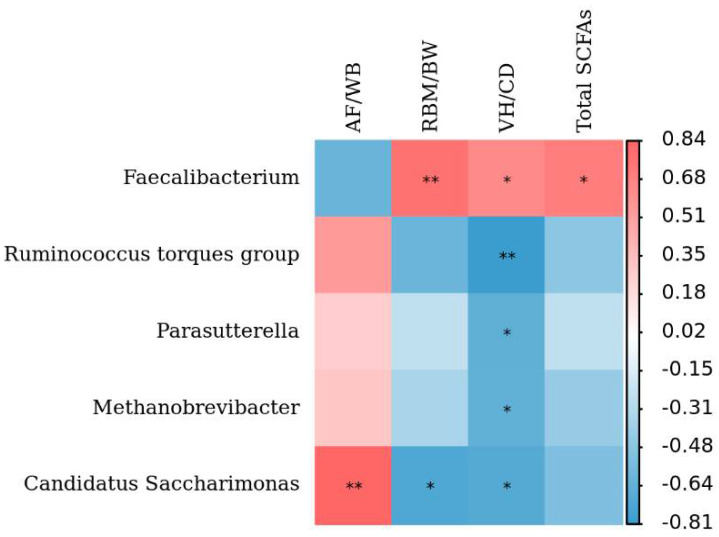
Heatmap of Spearman’s correlation analysis between the cecal microbiota composition and the growth performance indexes of geese. The red color represents a positive correlation, while the blue color represents an inverse correlation. ZG, Zi Goose; FG, Xianghai Flying Goose; AF/WB, abdominal fat index; RBM/WB, right breast muscle index; (abdominal fat index and right breast muscle index were defined as the abdominal fat mass and right breast muscle weight divided by the body weight, respectively); VH/CD, villus height/crypt depth; Total SCFAs, total short-chain fatty acids. * *p* < 0.05, ** *p* < 0.01.

**Table 1 animals-12-01815-t001:** Ingredients of the commercial diets provided to the geese at different stages.

Ingredients	Weeks	
	0–4	5–20
Apparent metabolic energy (MJ/kg)	11.2	10.85
Crude protein (%)	19.25	17.16
Crude fiber (%)	4.75	5.88
Crude ash (%)	5.13	5.08
Calcium (%)	0.80	0.80
Total phosphorus (%)	0.42	0.37
Total lysine (%)	0.90	0.65
Total methionine (%)	0.42	0.35

**Table 2 animals-12-01815-t002:** Breast muscle physical properties in the ZG and FG.

Item	ZG (*n* = 30)	FG (*n* = 30)	SEM ^1^	*p* Value ^2^
pH	6.10 ^a^	5.62 ^b^	0.12	0.003
Meat color	84.23 ^b^	89.76 ^a^	1.21	0.001
Shear force (N)	41.31 ^a^	36.48 ^b^	2.31	0.047
Water loss (%)	8.95 ^b^	20.81 ^a^	2.75	0.002
Cooking loss (%)	22.80	21.74	1.98	0.603

^a,b^ Means with different letters within the same row differ at *p* < 0.05. ZG, Zi goose; FG, Xianghai flying goose. ^1^ SEM: standard error of the mean. ^2^ Level of significance.

**Table 3 animals-12-01815-t003:** Breast muscle proximate composition (IMF content and fatty acid proportion) in the ZG and FG.

Items	ZG (*n* = 30)	FG (*n* = 30)	SEM ^1^	*p* Value ^2^
IMF (%)	2.98 ^a^	2.61 ^b^	0.10	0.001
Fatty acid proportions (% of total fatty acids)				
C14:0	0.25	0.22	0.01	0.138
C16:0	20.95 ^a^	20.08 ^b^	0.28	0.002
C16:1	2.06	2.24	0.11	0.53
C18:0	9.26 ^b^	10.59 ^a^	0.32	<0.0001
C18:1n9c	37.78 ^a^	32.87 ^b^	0.79	<0.0001
C18:2n6c	22.82 ^b^	24.98 ^a^	0.46	<0.0001
C18:3n3	0.50	0.46	0.03	0.067
C20:3n6	0.30	0.26	0.05	0.453
C20:4n6	6.09 ^b^	8.23 ^a^	0.41	<0.0001
C22:6n3	0.42 ^b^	0.49 ^a^	0.04	0.031
SFAs	30.32	30.76	0.31	0.159
MUFAs	39.86 ^a^	35.11 ^b^	0.84	<0.0001
PUFAs	29.82 ^b^	34.12 ^a^	0.76	<0.0001

^a,b^ Means with different letters within the same row differ at *p* < 0.05. ZG, Zi goose; FG, Xianghai flying goose; SFAs, saturated fatty acids; MUFAs, monounsaturated fatty acids; PUFAs, polyunsaturated fatty acids. ^1^ SEM: standard error of the mean. ^2^ Level of significance.

**Table 4 animals-12-01815-t004:** ASVs and cecal microbial alpha diversity, including the Chao1, Shannon, and Simpson indexes, for the ZG and FG.

Items	ZG (*n* = 5)	FG (*n* = 5)	SEM ^1^	*p* Values ^2^
Observed ASVs	665	637	51.41	>0.999
Chao1	664.40	636.85	51.52	>0.999
Shannon	7.82	7.46	0.26	0.310
Simpson	0.99	0.98	0.004	0.087

ZG, Zi goose; FG, Xianghai flying goose. ^1^ SEM: standard error of the mean. ^2^ Level of significance.

## Data Availability

The raw amplicon sequence data are available in the NCBI sequence read archive (SRA) under the accession number PRJNA858583.

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
