# Peer review of "Comparative Analyses of Production Performance, Meat Quality, and Gut Microbial Composition between Two Chinese Goose Breeds"

_animals, 2022, doi:10.3390/ani12141815_

Round 1

Reviewer 1 Report

Comparative Analyses of Production Performance, Meat Quality and Gut Microbial Composition between Two Chinese Goose Breeds

The manuscript describes important problem related to composition of the gut microflora in geese, its influence on fermentation and amount of SCFA  formed in the caecum in two breeds of Chinese geese: Zi Goose and Xianghai Flying Goose. The text is legible and interesting. Has a  large  importance in terms of the selection of the goose  breed on the Chinese market, which is responsible for most of the goose meat produced in the world, hence the increase amount of PUFA in meat affects the health benefits  of such product, which in the long run also reduces the occurrence of civilization diseases related to fatty acid changes in the organism, especially the level of LDL. However, before its publication, it is needed pay attention to a few details described below: 

Line 49: Space before Yan Zhang, it  should be corrected

Line 163:  Word repetition in one sentence:’ …breast muscle yield in poultry ‘ , could be change for breast muscle yield in birds

Line 247: Information about lighting programme is needed in week 0-4 (normally 24h in first days, then decreased to environmental in week 8 )

Line 250: 25-26 degrees in first week then it is decreased about 2 degrees in a week. Maybe better to describe that: temperature in room obtain 25-26 in first day of breeding, and then were decreased to 20 degrees in week 4 and finally was the same like in environmental in week 8

Line 251: There exist a risk to treat 1 pen only as one repetition of a mean value in each treatment. It will better to separate whole pen with 30 birds for 10 pens with 3 geese, that would be more representative due to different microclimate in each pen and in each part of pen or mixed same birds in two pens (30 of each, 50% M:50% F, from two different lines on the same mixtures) with bands in two different colours to identify birds from different groups.

Line 256: MJ/kg instead of KJ/kg

Line 259: birds were sacrificed by cervical dislocation (no. approval of ethics committee could be also useful information)

Line 351: Water loss:  better is to present as an equation:

Line 434 and 439: Maybe is better to merge subsections 2.7 and 2.8 as a Statistical analysis. In consequence in first paragraph could be presented information about using Student’s t-test as main tool in statistical analysis and in second paragraph Spearman correlation describes relation between cecum microbiota and performance indexes

Line 446: Information about exact value in brackets will be useful, especially in case of F and H chart, ie. villus height, there is more than 600 µm in case of ZG and 700 something looking for FG breed geese, it is difficult to interpret (or add values on chart)

Line 479: In whole third chapter (Results) is needed to add values in text between comparison of different estimators

Line 663: Number of article should be used after year of publication in text: Weng et al. 2022 [20]

Line 788: ‘Different goose breeds have different cecal microbiota compositions, and different cecal microbiota compositions contribute to different production performance traits.’ Synonyms of different needed (ie. varied, diverse,various, …)

Line 809: et al. could be use only in text of article in references all authors are must be presented.

Author Response

Response:

Line 49: Space before Yan Zhang, it should be corrected.

Response: Thank you for bringing this issue to our attention. We have revised this in our manuscript.

Line 163:  Word repetition in one sentence:’ …breast muscle yield in poultry ‘ , could be change for breast muscle yield in birds

Response: Thanks, we have replaced “breast muscle yield in poultry” with “breast muscle yield in birds” in lines 55-56.

Line 247: Information about lighting programme is needed in week 0-4 (normally 24h in first days, then decreased to environmental in week 8)

Response: Thank you for your advice. We have detailed the lighting scheme for goose breeding conditions as follows:

Lines 100-103 “Sixty goslings were kept in a shed to ensure stable rearing conditions and then gradually transferred to the outside. The first day of breeding was under 24 h of light and gradually decreased to natural light conditions in week 8.”.

Line 250: 25-26 degrees in first week then it is decreased about 2 degrees in a week. Maybe better to describe that: temperature in room obtain 25-26 in first day of breeding, and then were decreased to 20 degrees in week 4 and finally was the same like in environmental in week 8.

Response: Thank you for your suggestions. We have revised our manuscript as follows:

Lines 103-106 “The indoor temperature ranged from 25-26°C in the first week of breeding, then decreased to 20°C in week 4 (approximately 2°C a week), and finally was the same as the ambient temperature in week 8.”

Line 251: There exist a risk to treat 1 pen only as one repetition of a mean value in each treatment. It will better to separate whole pen with 30 birds for 10 pens with 3 geese, that would be more representative due to different microclimate in each pen and in each part of pen or mixed same birds in two pens (30 of each, 50% M:50% F, from two different lines on the same mixtures) with bands in two different colours to identify birds from different groups.

Response: Thank you for your suggestions. We have revised our manuscript as follows:

Lines 106-108 “The two breeds of geese were raised in two separate pens (20×30 m), with 30 birds per pen, for a total of 2 pens, and each pen was divided into 5 small pens with 6 geese in each small pen.”

Line 256: MJ/kg instead of KJ/kg

Response: Thank you for bringing this issue to our attention. We have revised KJ/kg to MJ/kg in Table 1.

Line 259: birds were sacrificed by cervical dislocation (no. approval of ethics committee could be also useful information)

Response: Thank you for your common. The approval of the ethics committee is mentioned in the Institutional Review Board Statement section:

Lines 407-409. “All animal experimental procedures were approved by the Institutional Animal Care and Use Committee of Jilin University, China (SY202105020).”

Line 351: Water loss:  better is to present as an equation:

Response: Thank you for your comment. The equation is mentioned in our manuscript as follows:

Lines 153-157 “Approximately 1 g (W1) of muscle was weighed, and 10 layers of filter paper were placed on the top and bottom of the sample. Then, the covered sample was placed on the dilatometer platform for 5 minutes at a pressure of 68.66 kPa, and the weight of the muscle sample was measured again (W2) to calculate the amount of released water as follows: Water loss (%) = (W1-W2)/W1 × 100%.”

Line 434 and 439: Maybe is better to merge subsections 2.7 and 2.8 as a Statistical analysis. In consequence in first paragraph could be presented information about using Student’s t-test as main tool in statistical analysis and in second paragraph Spearman correlation describes relation between cecum microbiota and performance indexes

Response: Thank you for your suggestion. We have merged subsections 2.7 and 2.8 into one part in lines 213-221.

Line 446: Information about exact value in brackets will be useful, especially in case of F and H chart, ie. villus height, there is more than 600 µm in case of ZG and 700 something looking for FG breed geese, it is difficult to interpret (or add values on chart)

Response: Thank you for your advice. We have revised this in in the text.

Line 479: In whole third chapter (Results) is needed to add values in text between comparison of different estimators.

Response: Thanks, we have added values in the text between the comparison of different estimators.

Line 663: Number of article should be used after year of publication in text: Weng et al. 2022 [20]

Response: Thank you for bringing this issue to our attention. We have moved the number of articles after the year of publication in our manuscript in line 312.

Line 788: ‘Different goose breeds have different cecal microbiota compositions, and different cecal microbiota compositions contribute to different production performance traits.’ Synonyms of different needed (ie. varied, diverse,various, …)

Response: Thank you for your comment. We have revised it in our manuscript as follows:

Lines 391-392: “Different goose breeds have diverse cecal microbiota compositions, and various cecal microbiota compositions contribute to dissimilar production performance traits”.

Line 809et al. could be use only in text of article in references all authors are must be presented.

Response: Thank you for your advice. We have revised the format of the references.

Reviewer 2 Report

Comments:

"Goose meat is preferred by consumers because it"; preferred to what? It is not the preferred meat in my opinion, I would use the word consumed instead of preferred

Remove underscores from taxa names, they are created in R to avoid spaces, valid taxa do not have underscores like in Candidatus_Saccharimonas and elsewhere

Is it NovaSeq 6000?

QIIME (Version 1.7.0) was released a long time ago in May 2013! Qiime 2 was introduced in 2018 and qiime 1 is fully depreciated. OTU concept is all but abandoned, the data analysis is quite questionable

Were the data chimera checked and denoised?

Did you use Silva or GreenGenes for taxonomy?

Two-tailed Student’s t-test is not suitable for microbiota stats, it is not normally distributed. Was the data rarefied or normalised?

You cannot make valid comments on breed-to-breed differences based on one trial, this needs to be specified "in this particular trial differences ..." or in a similar way

The difference between groups cannot be claimed based on a few individual univariate differences, multivariate analysis such as permanova or adonis can give us insight if the groups are different or not.

The data analysis is outdated and not done properly (Chimera, denoising, silva taxonomy etc are needed), it needs to be re-analysed and more analysis details added to the methods section. 

Author Response

Response:

"Goose meat is preferred by consumers because it"; preferred to what? It is not the preferred meat in my opinion, I would use the word consumed instead of preferred

Response: Thank you for your suggestion. We have replaced “preferred” with “consumed”.

Line 21 “Goose meat is consumed…”.

Line 29 “Goose meat is consumed…”.

Remove underscores from taxa names, they are created in R to avoid spaces, valid taxa do not have underscores like in Candidatus_Saccharimonas and elsewhere

Response: Thank you for bringing this to our attention. We have removed the underscores from the taxon names.

Is it NovaSeq 6000?

Response: The NovaSeq 6000 platform was used in our experiments.

Line 185-186: “The 250-bp paired-end amplicon libraries were sequenced using the Illumina NovaSeq 6000 platform.”

QIIME (Version 1.7.0) was released a long time ago in May 2013! Qiime 2 was introduced in 2018 and qiime 1 is fully depreciated. OTU concept is all but abandoned, the data analysis is quite questionable

Response: Thank you for your advice. QIIME (Version 1.9.1) was used for our data analysis, and it was written as QIIME (version 1.7.0) due to our negligence. The main reason why we chose QIIME 1 for data analysis is that we previously completed some relevant studies using QIIME1 for data analysis. Therefore, to compare with previous data, QIIME 1 was selected for our data analysis. In addition, we searched the relevant literature and found that although QIIME 2 replaced QIIME 1 on January 1, 2018, QIIME 1 data analysis continued to be used for nearly 2 years. For example, an article published in Science in 2020 entitled “Multi-omics analyses of radiation survivors identify radioprotective microbes and metabolites” and an article published in Poultry Science in 2021 entitled “The differences in intestinal growth and microorganisms between male and female ducks”.

Were the data chimera checked and denoised?

Response: The data were chimera checked and denoised. We have added some sentences in our manuscript as follows:

Lines 186-192 “QIIME software (Version 1.9.1) was used to remove the barcodes, primers, and low-quality sequences with a Q < 20, and FLASH (VI.2.7) [17] was used to merge high-quality paired-end clean reads into tags [18]. Then, the tags were compared with the Silva database using the UCHIME algorithm to remove chimera sequences and obtain effective sequences [19]. Finally, on average, 64,230 and 66,565 effective sequences resulted from each ZG and FG cecal content sample, respectively, for further analysis. The obtained effective…”

Did you use Silva or GreenGenes for taxonomy?

Response: We used the Silva database for taxonomy, and it is mentioned in line 194.

Two-tailed Student’s t-test is not suitable for microbiota stats, it is not normally distributed. Was the data rarefied or normalised?

Response: Thank you for your comment. Two-tailed Student’s t test was used for production performance and meat quality data analysis, and the Wilcoxon rank sum test and nonparametric Kruskal–Wallis test were used for microbiota data analysis. We have revised it in our manuscript as follows:

Lines 217-219: “Analyses were performed using GraphPad Prism software version 8 and software R (V.2.15.3). Data were tested using unpaired two-tailed Student’s t tests Wilcoxon rank sum test, and nonparametric Kruskal–Wallis test.”

You cannot make valid comments on breed-to-breed differences based on one trial, this needs to be specified "in this particular trial differences ..." or in a similar way

Response: Thank you for your suggestion. We have revised our manuscript as follows:

Line 45 “Taken together, in this particular trial, FG had better…”.

Line 393 “In this certain trial, the results showed that FG had a lower…”.

The difference between groups cannot be claimed based on a few individual univariate differences, multivariate analysis such as permanova or adonis can give us insight if the groups are different or not.

Response: Thank you for your comment. There was an analysis of PCoA in our manuscript in Figure 3.

The data analysis is outdated and not done properly (Chimera, denoising, silva taxonomy etc are needed), it needs to be re-analysed and more analysis details added to the methods section.

Response: Thank you for your advice. We have added some analysis details in the methods section as follows:

Lines 186-203 “…QIIME software (Version 1.9.1) was used to remove the barcodes, primers, and low-quality sequences with a Q < 20, and FLASH (VI.2.7) [17] was used to merge high-quality paired-end clean reads into tags [18]. Then, the tags were compared with the Silva database using the UCHIME algorithm to remove chimera sequences and obtain effective sequences [19]. Finally, on average, 64,230 and 66,565 effective sequences resulted from each ZG and FG cecal content sample, respectively, for further analysis. The obtained effective sequence fragments were clustered into operational taxonomic units (OTUs) with a similarity threshold of 97% and assigned taxonomies to representative sequences based on the Silva database [20] using the QIIME (Version 1.9.1) software package. The OTUs were analyzed for alpha diversity (Ace, Chao1, Shannon, and Simpson indexes) [21] using QIIME (Version 1.9.1). Beta diversity analysis based on the Bray–Curtis distance was generated using QIIME (Version 1.9.1) and displayed by principal coordinate analysis (PCoA) in the R language. A heatmap was generated based on phylum and genus information using the R function heatmap, and the significant difference between the ZG and FG was evaluated using the Benjamini‒Hochberg corrected p value (false discovery rate < 0.05). The metabolic capabilities of the cecal microbiota were analyzed based on the FAPROTAX database using 16S rRNA sequencing data….”. However, due to the revision time constraints, we did not reanalyze our data using QIIME2.

Round 2

Reviewer 2 Report

The authors attempted to answer and address the questions raised by reviewers. I am not quite sure what is going on, qiime 1.7 was a mistake it was actually qiime 1.9, and t-test was actually Wilcoxon etc. 

The argument that the authors used qiime 1.9 because the old data is done using this software is a poor excuse for using outmoded algorithms for microbiota analysis. Comparing themselves with some other papers that got away with using qiime 1.9 is also very poor practice. The data was not denoised and not properly analysed. Qiime 2 was generated as a new software rather than an update after major issues with the data analysis were detected and corrected in qiime 2.

The authors need to keep up with the progress in the area. I would suggest reanalysing that data with data denoise

I will not select reject and leave this up to the editor to decide but I suggest the authors should upgrade the algorithms if they want to keep publishing in the area 

Author Response

Thank you for your suggestion. We have reanalyzed our data using QIIME2 and revised our manuscript in the Abstract, 2.5. DNA Extraction, Microbiota Analysis, and Functional Prediction, 2.7. Statistical Analysis, 3.3. Comparison of Cecum Microbiota Composition, Functional Prediction and SCFA Concentrations between the Two Goose Breeds, 3.4. Associations of the Cecum Microbiota Composition with Production Performance Indexes, and 4. Discussion section. In additon, we have revised Table 4, Figure 3, Figure 4, and Figure 6.